

# Snell's Law for a vortex dipole in a Bose-Einstein condensate

Michael M. Cawte[1], Xiaoquan Yu[1], Brian P. Anderson[2] and Ashton S. Bradley[1⋆]

**1** Dodd-Walls Centre, Department of Physics, University of Otago, New Zealand
**2** College of Optical Sciences, University of Arizona, Tucson AZ 85721, USA

⋆ ashton.bradley@otago.ac.nz

## Abstract

A quantum vortex dipole, comprised of a closely bound pair of vortices of equal strength with opposite circulation, is a spatially localized travelling excitation of a planar superfluid that carries linear momentum, suggesting a possible analogy with ray optics. We investigate numerically and analytically the motion of a quantum vortex dipole incident upon a step-change in the background superfluid density of an otherwise uniform two-dimensional Bose-Einstein condensate. Due to the conservation of fluid momentum and energy, the incident and refracted angles of the dipole satisfy a relation analogous to Snell's law, when crossing the interface between regions of different density. The predictions of the analogue Snell's law relation are confirmed for a wide range of incident angles by systematic numerical simulations of the Gross-Pitaevskii equation. Near the critical angle for total internal reflection, we identify a regime of anomalous Snell's law behaviour where the finite size of the dipole causes transient capture by the interface. Remarkably, despite the extra complexity of the interface interaction, the incoming and outgoing dipole paths obey Snell's law.

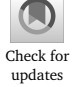

# 1  Introduction

One of the defining characteristics of superfluids is their ability to support quantized vortices that carry angular momentum. A singly quantized vortex is a topological defect in the macroscopic wavefunction of the superfluid in which the phase of the wavefunction winds around a core of zero density. In a Bose-Einstein condensate (BEC) under planar confinement, vortex bending is suppressed and vortex motion can become effectively two dimensional (2D) [1]. 2D quantum vortex systems support a rich phenomenology [2], including vortex clusters [3], the Kosterlitz-Thouless phase [4, 5], and negative-temperature states [6–13]. A vortex closely bound with a vortex of opposite circulation (an antivortex) in a BEC forms a vortex dipole that carries linear fluid momentum [14]; these spatially localized excitations propagate with a constant speed which is inversely proportional to the distance between the vortices. Vortex dipoles have been created in BECs confined by parabolic potentials [15, 16], and could potentially be precisely manipulated using blue and red-detuned laser beams [17]. They play a central role in the breakdown of superfluidity [18–20] and in energy transport mechanisms underpinning 2D quantum turbulence (2DQT) [21–25]. Vortex dipoles have also been observed in strongly damped exciton-polariton systems [26].

Advances in spatiotemporal optical trap control in BECs [27–29] now enable a broad range of superfluid dynamics experiments, and the possibility of studying detailed vortex motion is increasingly within reach of current technology. In recent numerical work [30], it was shown that a vortex dipole at sufficiently high velocity could cross an interface in an immiscible two-component BEC. At lower velocities, the dipole either disappeared or disintegrated, with the remnants moving along the interface. For vortex dipoles with an oblique angle of incidence upon the interface, the cores of the dipole were found to be asymmetrically filled with the first component of the BEC after propagation into the second component. In general, the dipole-interface interaction was found to be quite complex.

In this work, we study the motion of a single vortex dipole incident on a density-step interface in a single-component BEC. The interface is created by adding an abrupt potential step, resulting in two large regions of different potential and density. We numerically solve the Gross-Pitaevskii equation (GPE) describing the trapped BEC and systematically study the transmission and reflection characteristics of the dipole as a function of its angle of incidence and initial dipole momentum. Accounting for fluid momentum and energy conservation of the dipole in the two homogeneous regions, the incoming and outgoing trajectories of the dipole are found to obey an analogous form of Snell's law. Formulating the correct Snell's law suggests a means of understanding vortex dipole motion in inhomogeneous superfluids based on optical analogies.

In optics, Snell's law states that the ratio of the sines of the incident and outgoing angles $(\theta_i, \theta_f)$ of a ray of light traversing through two media is given by the reciprocal ratio of the

indices of refraction for each medium:

$$\frac{\sin\theta_f}{\sin\theta_i} = \frac{n_i}{n_f}; \qquad \text{where} \quad n_i \equiv \frac{c_v}{v_i}, \quad n_f \equiv \frac{c_v}{v_f}. \tag{1}$$

The index of refraction of a medium is the ratio of the speed of light in the vacuum ($c_v$) to the phase velocity of light in the medium ($v_i, v_f$) (1).

The natural reference speed for defining the refractive index for a superfluid vortex dipole is the speed at which the vortices lose their topological character and coalesce into a localised density and phase excitation known as the *Jones-Roberts soliton* (JRS) [31]; this speed is less than the speed of sound in each region of the system and sets a natural boundary to the parameters of the dipoles we study, in particular setting the scale of the smallest dipoles that we may consider.

The structure of this paper is as follows. In Section 2 we give a brief background on the Gross-Pitaevskii equation for BEC dynamics and quantum vortex dipoles. In Section 3 we present the form of Snell's law applicable to quantum vortex dipoles in a compressible 2D superfluid. In Section 4 we present our numerical simulations of vortex dipole motion across a step interface in the superfluid density. In Section 5 we discuss the link between the JRS regime and the optical concept of a refractive index and present our conclusions.

# 2 Background

## 2.1 Gross-Pitaevskii equation

The Gross-Pitaevskii equation (GPE) provides a reliable description of BEC dynamics far below the critical temperature, $T \ll T_c$ [32]. We consider a BEC state described by wavefunction $\Psi(\mathbf{r}, t)$ normalized to the total number of BEC atoms

$$\int d^3\mathbf{r} \, |\Psi(\mathbf{r}, t)|^2 = N. \tag{2}$$

In three dimensions, the BEC energy is

$$E = \int d^3\mathbf{r} \, \Psi^*(\mathbf{r}, t) \left( -\frac{\hbar^2 \nabla^2}{2m} + V_{\text{ext}}(\mathbf{r}, t) \right) \Psi(\mathbf{r}, t) + \frac{g}{2} |\Psi(\mathbf{r}, t)|^4, \tag{3}$$

and the equation of motion is given by the GPE

$$i\hbar \frac{\partial \Psi(\mathbf{r}, t)}{\partial t} = \left( -\frac{\hbar^2}{2m} \nabla^2 + V_{\text{ext}}(\mathbf{r}, t) + g|\Psi(\mathbf{r}, t)|^2 \right) \Psi(\mathbf{r}, t), \tag{4}$$

where $\hbar$ is the reduced Planck constant, $V_{\text{ext}}$ is the external potential, $g = 4\pi\hbar^2 a_s/m$ is the two-body interaction parameter, $m$ is the atomic mass and $a_s$ is the s-wave scattering length. We assume the system is tightly confined in the $z$ direction so that the wavefunction is separable $\Psi(\mathbf{r}, t) \equiv \psi(x, y, t)\psi_0(z)$. Integrating over the $z$ direction, the GPE can be written in terms of an effective 2D interaction parameter $g_{2D} = (4\pi\hbar^2 a_s/m) \int |\psi_0|^4 dz$ where $\int d^2\mathbf{r} \, |\psi_0|^2 = N$. Defining the healing length

$$\xi \equiv \frac{\hbar}{\sqrt{m\mu}}, \tag{5}$$

where the chemical potential $\mu = g_{2D}\rho_0$ is related to $\rho_0$, the homogeneous 2D atomic number density in the absence of an $x$-$y$ trapping potential. For the homogeneous system, convenient units of length, time and energy are $\xi$, $\xi/c$ and $\mu = \hbar^2/m\xi^2$ respectively, where $c \equiv \sqrt{g_{2D}\rho_0/m}$ is the speed of sound.

## 2.2 Vortex dipoles

Vortex dipoles play a fundamental role in the dynamics of two-dimensional superfluids. Whereas a single vortex carries angular momentum and has an excitation energy logarithmically dependent on the size of the system, a vortex dipole has an excitation energy that only depends on the separation distance between the pair of vortices. Here and in the remainder of our paper, the vector $\mathbf{r}$ is the two-dimensional cartesian vector $(x, y)$. We adopt the ansatz for the core of a single vortex $\sqrt{\rho(r)} = \sqrt{\rho_0} f(r)$ where [33]

$$f(r) = \frac{r}{\sqrt{r^2 + \xi^2}}, \tag{6}$$

and $r$ is the distance from the centre of the vortex core. The phase of a single vortex of circulation $\kappa_j$ is given by $\theta_j(x, y) = \kappa_j \text{atan2}(x - x_j, y - y_j)$, where atan2 is defined by

$$\text{atan2}(x, y) = \begin{cases} \arctan(y/x), & x > 0 \\ \arctan(y/x) + \pi, & x < 0, y > 0 \\ \arctan(y/x) - \pi, & x < 0, y < 0. \end{cases} \tag{7}$$

This choice causes the branch cut for a single vortex to lie along the negative $y$-axis.

Assuming no fluid boundaries are nearby and the vortices are well-separated, the wavefunction of a vortex dipole can be similarly constructed as

$$\psi(\mathbf{r}) = \sqrt{\rho_0} f(\mathbf{r} - \mathbf{r}_+) f(\mathbf{r} - \mathbf{r}_-) e^{i\varphi(\mathbf{r})}, \tag{8}$$

where the phase is

$$\varphi(\mathbf{r}) = \text{atan2}(x - x_+, y - y_+) - \text{atan2}(x - x_-, y - y_-), \tag{9}$$

and $\mathbf{r}_\pm \equiv (x_\pm, y_\pm)$ is the vortex position with circulation $\pm 1$.

We can now calculate the fluid momentum

$$\mathbf{P} = \int d^2\mathbf{r}\, \psi^*(-i\hbar\nabla)\psi \tag{10}$$

for the vortex dipole with inter-vortex separation $d = |\mathbf{r}_+ - \mathbf{r}_-| \gg \xi$, using (8) and neglecting the small contribution from the density gradient near the vortex cores. First, we note that there is a phase difference of $\pi$ between any two points on either side of a single vortex. Without loss of generality, we consider a dipole with vortices at $\mathbf{r}_\pm = (0, \pm d/2)$. For our chosen dipole we have $P_y = 0$ (by symmetry), and

$$P_x = \hbar\rho_0 \int_{-L/2}^{L/2} dy \int_{-L/2}^{L/2} dx\, \partial_x \varphi(\mathbf{r}) = \hbar\rho_0 \int_{-d/2}^{d/2} dy\, \Delta\varphi_L(y), \tag{11}$$

where by neglecting density-gradient terms we reduce the integration along $x$ to evaluating the phase difference $\Delta\varphi_L(y) \equiv \varphi(L/2, y) - \varphi(-L/2, y)$ across the dipole in the $x$-direction. The phase difference is zero for $|y| > d/2$. Inside the interval $|y| < d/2$ the phase difference is always $2\pi$ when $L$ is sufficiently large (strictly when $L \to \infty$, but the limit is rapidly approached for $L \gg d$). The magnitude of the quantum vortex dipole momentum is thus given by [32]

$$P \equiv |\mathbf{P}| = 2\pi\hbar\rho_0 d. \tag{12}$$

The excitation energy of a vortex dipole $E_d$ can be obtained by inserting Eq. (8) into the two-dimensional expression for the Gross-Pitaevskii energy relative to a uniform background with density $\rho_0$ [33]

$$E = \int d^2\mathbf{r}\, \frac{\hbar^2|\nabla\psi|^2}{2m} + \frac{g_{2D}}{2}(|\psi|^2 - \rho_0)^2,$$ (13)

giving the result

$$E_{\mathrm{d}} = 2\pi\rho_0 \frac{\hbar^2}{m}\left[\ln\left(\frac{d}{\xi}\right) + \frac{1}{4} + \frac{1}{2}\right].$$ (14)

Here the first term is the kinetic energy of the fluid in the hydrodynamic limit, which depends on the dipole separation distance only and does not scale with the system size, as far from the vortex dipole the velocity field vanishes. The second and third terms are the contributions from the quantum pressure and the interaction energy change due to atomic density depletion relative to the homogeneous background. As shown by Fetter [33], the core correction to the fluid momentum is negligible for $d \gtrsim \xi$, while the weaker logarithmic dependence of the energy on $d$ necessitates a careful account of the density gradient contributions from the vortex core.

For the exact numerical solution the vortex dipole energy can be written in a similar form:

$$E_{\mathrm{d}} = 2\pi\rho_0 \frac{\hbar^2}{m}\ln\left(\frac{\alpha d}{\xi}\right),$$ (15)

where the parameter $\alpha \simeq 2.07$ is numerically found to be very close to the ansatz result $e^{(1/4+1/2)} \simeq 2.117$, and accurately accounts for the quantum pressure and interaction energy. We use the numerically precise value $\alpha$ to accurately account for the quantum pressure and interaction energy changes incurred when a dipole moves between regions of superfluid with different background densities. Finally, we note that, irrespective of the precise treatment of the core energy, the speed of the dipole for a given dipole moment $d$ may be found via

$$v_{\mathrm{d}} = \frac{\partial E_{\mathrm{d}}}{\partial p} = \frac{\hbar}{md}.$$ (16)

We emphasize that in contrast with intuition based on the Newtonian mechanics of classical particles, the dipole energy and momentum scale with $d$: larger $d$ gives larger dipole energy and momentum, but smaller dipole velocity.

## 3  Snell's Law for vortex dipoles

We consider a superfluid with densities $\rho_1$ and $\rho_2$ separated by a sharp interface in the density created by an external potential (see Fig. 1). There are two healing lengths dependent on the local chemical potential $\xi_j = \hbar/\sqrt{mg_{2D}\rho_j}$ for $j = 1, 2$. Hereafter, we work in units of the healing length $\xi = \min(\xi_1, \xi_2)$, the shortest length scale set by the highest particle density. To distinguish the static background densities $\rho_j$ from dynamical properties of the vortex dipole, we use the subscripts $(i, f)$ to refer to initial and final states of the dipole respectively. The initial or final state may refer to a dipole in either region $\rho_1$ or $\rho_2$ of superfluid density, as will be made clear from the context.

We prepare a vortex dipole far from the interface, characterized by the initial separation $d_i$ and fluid momentum with initial magnitude $P_i \equiv |\mathbf{P}_i| = 2\pi\hbar\rho_i d_i$ and energy as $E_i =$

$(2\pi\rho_i\hbar^2/m)\ln(\alpha d_i/\xi_i)$. We refer to this as the asymptotic *incoming* state. After interacting with the interface, the asymptotic *outgoing* state is characterized by the corresponding final quantities $d_f$, $P_f = 2\pi\hbar\rho_f d_f$ and $E_f = (2\pi\rho_f\hbar^2/m)\ln(\alpha d_f/\xi_f)$. Conservation of the fluid momentum parallel to the interface gives

$$P_i\sin(\theta_i) = P_f\sin(\theta_f). \qquad (17)$$

When the vortex dipole crosses the interface, in general, some incompressible fluid energy can transfer to acoustic energy due to the change of the vortex core size. In this analysis, we assume that the energy transfer is negligible, as expected when the change in density is small, and we later verify our assumption with systematic GPE simulations. Provided this assumption is valid, we have the conservation law

$$E_i = E_f. \qquad (18)$$

Combing Eq. (17) and Eq. (18), the angle and the separation of the outgoing vortex dipole can be expressed as

$$\theta_f = \arcsin\left(\sin\theta_i\frac{P_i}{P_f}\right), \qquad (19a)$$

$$d_f = \frac{\xi_f}{\alpha}\left(\frac{\alpha d_i}{\xi_i}\right)^{\frac{\rho_i}{\rho_f}}. \qquad (19b)$$

The formulation of Snell's law for vortex dipoles given in Eqs. (19) provides the main analytical result of our paper; in the remainder of the paper, our aim is to test this formulation of Snell's law by numerically solving the Gross-Pitaevskii equation. For $\rho_1 < \rho_2$, when the incident angle reaches a critical value

$$\theta_c \equiv \arcsin\left(\frac{P_f}{P_i}\right), \qquad (20)$$

the angle of refraction approaches 90°. This is the expected behaviour for an *ideal* Snell's-law process, where the structure of the dipole or interface are not important.

In optics, Snell's law approximates the propagation of electromagnetic waves in different materials, when the wavelength is much smaller than the other relevant scales. In our case, the motion of the dipole represents a complex flow pattern of the underlying superfluid due to the motion of the constituent quantum vortices. If we instead view a vortex dipole as a particle, discarding the underlying fluid motion, a naive application of Fermat's principle leads to an incorrect refraction formula: $\sin\theta_i/\sin\theta_f = v_i/v_f$, where $v_j = \hbar/(md_j)$ is the velocity of a vortex dipole. Our results establish that a quantum vortex dipole describing a collective excitation of the host superfluid obeys a Snell's law type relation, and hence its motion bears a close analogy to the rectilinear propagation of optical rays between media of constant refractive index.

For quantum vortex dipoles, there is a well-understood short-range phenomenology occurring as the vortices approach each other closely. The process was studied theoretically by Jones and Roberts [31], who established that once the vortices reach a critical separation, they lose their exact integer phase winding, and are no longer topological excitations of the superfluid. The two cores partially merge to form a single propagating region of low-density, and the vortex dipole gives way to a localised excitation known as the *Jones-Roberts soliton* [31]. The onset of the JRS regime can be estimated as the dipole speed at which the fluid velocity in the middle of a vortex dipole (the highest velocity) reaches the speed of sound. This immediately yields $d_{JRS} \approx 3\xi$, consistent with the numerical calculation that yields $d_{JRS} \approx 2.3\xi$ [31]. In this paper, we only consider the situations where $d \gg d_{JRS}$, and the vortices remain point-like.

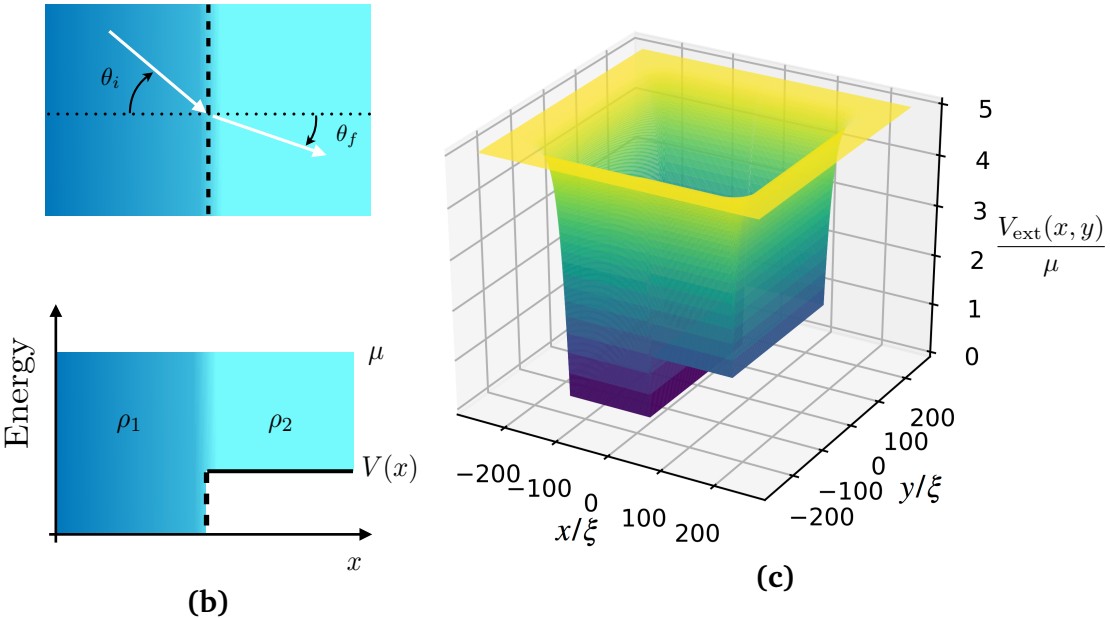

Figure 1: **(a)**: Schematic top-down perspective of a superfluid with a sudden change in density producing an interface. The dipoles approach the interface at incident angle $\theta_i$ and leave at an angle of refraction $\theta_f$ with respect to the normal of the interface. **(b)**: Cross-section perspective of the superfluid in figure **(a)**. The fluid density is reduced after a step increase in the external potential. **(c)**: Surface plot of the external potential $V_{\text{ext}}$, which has dimensions $512\xi \times 512\xi$. A buffer of $100\xi$ on each side prevents numerical noise in simulations. The potential well features an adjustable step allowing for control of the superfluid density in one half of the system.

## 4 Numerical Simulations

In this section, we present our numerical simulations of vortex dipole propagation in a superfluid described by the GPE. All simulations were run by using the Julia programming language [34]. The GPE was solved by using pseudo-spectral methods with the DifferentialEquations.jl package [35] and Tsit5 [36] algorithm. Vortices were detected by using the VortexDistrubutions.jl package [37].

### 4.1 Initial Conditions

To set up an interface between two regions of a homogeneous BEC with different densities, we constructed an external potential $V_{\text{ext}}(\mathbf{r}, t)$ in the shape of a box trap with an adjustable step. This was created by specifying sharp masks for the interface and trapping boundaries, and smoothing these features by locally applying a $\tanh^6(x/\xi)$ to the interface mask and $\tanh^6(x/4\xi)$ to each boundary mask. In detail, the potential is taken to be of the form

$$V_{\text{ext}}(x) = 5\mu \tanh^6\left(\frac{w}{x + L/2 - \xi/2}\right) + h \tanh^6\left(\Theta(x)\frac{x}{w}\right) + (5\mu - h)\tanh^6\left(\frac{w}{x - L/2}\right), \quad (21)$$

for $-L/2 + \xi/2 < x < L/2$ and given $-L/2 + \xi/2 < y < L/2$, where $h = \mu - g_{2D}\rho_2$, $w = 4\xi$, and $\Theta(x)$ is the Heaviside step function. While for $|x| > L/2$ and $|y| > L/2$, $V_{\text{ext}} = 5\mu$. An example potential is shown in Figure 1c. We first find the ground state of the system using imaginary time evolution of the GPE under the constraint of fixed particle number. The incoming vortex

dipole is created by imprinting the phase Eq. (8) and the density Eq. (6) onto the ground state. After imprinting the vortex dipole, we apply a short imaginary time evolution to damp out the acoustic excitation energy induced by imprinting the ansatz Eq. (6) (since it is not the exact core solution). Once acoustic energy is removed, the dipole is evolved according to the Hamiltonian GPE. The incoming dipole distance $d_i \gg \xi_i$ is chosen large enough to ensure that the distance of the outgoing dipole $d_f \gg d_{\mathrm{JRS}}$[1]. For particular angles of incidence, we found more precision was required when setting up the initial dipole parameters. In this case, we imprinted the numerically exact vortex core, as described later in this article.

## 4.2 Representative Examples of Time Evolution

We first present atomic density plots showing the evolution of three representative examples of total internal reflection observed in our GPE simulations. The notable feature of the dipole motion is the possibility that the dipole can partially cross the interface before being reflected. The examples in Figure 2 (**a**), (**b**) and (**c**) each use an external potential step to create an interface between two regions ($\rho_1 \xi^2 = 8.5$, $\rho_2 \xi^2 = 10$) with the dipole moving from low to higher density. The initial dipole separations are $d_i = (17.9, 17.9, 17.8)\xi$. Recalling the critical angle formula (20) and using the initial dipole separation and fluid densities, the critical angles for the three dipoles are $\theta_c = (39.7°, 39.7°, 39.6°)$ respectively. In Figure 2 (**a**), a dipole incident at the interface just above the critical angle ($\theta_i = 41.4°$), undergoes transient transmission of both vortices into the second region, before reflecting back into the first region with the outgoing angle and dipole separation close to the incoming values. In Figure 2 (**b**), a dipole of initial separation $d_i = 17.9\xi$ is initialized in a fluid of density $\rho_1 \xi^2 = 8.5$ with incident angle $\theta_i = 58.7°$. As the dipole approaches the interface one of the vortices partially immerses into the denser region before the dipole is reflected with outgoing dipole separation and angle close to the incoming values. In simulations with larger incident angles, the dipole always undergoes total internal reflection, with neither vortex crossing into the denser region. In Figure 2 (**c**) a dipole of similar initial separation $d_i = 17.8\xi$ is initialized in a fluid of density $\rho_1 \xi^2 = 8.5$ with incident angle $\theta_i = 50.6°$, between the incident angles of Figure 2 (**a**) and (**b**). At the interface, only one vortex of the pair crosses into the denser region with the other remaining in the original region. The dipole spends an extended period of time straddling the interface before reflection at close to the initial dipole separation and incident angle.

The previous three figures show the range of behaviour for dipoles with incident angles above the critical angle at a sharp line interface with a region of higher density. An elongated confining geometry had to be used in Figure 2 (**c**) to capture the full dynamics of dipole trajectories that spend an extended period of time straddling the interface and ensure well-defined asymptotic states that are not perturbed by the system boundary.

## 4.3 Low- to High-Density Trajectories

We now consider a range of values for $\rho_1/\rho_2$. We set the fluid density of the first region so that $\rho_1 < \rho_2$, and study the relationship between changes in fluid density and critical/refracted angles. Three separate external potentials were used to set up changes in superfluid density, that we use in the rest of this article. The parameters for each case are referred to as

1. **Small:** $\rho_1/\rho_2 = 9.5/10$

2. **Medium:** $\rho_1/\rho_2 = 9/10$

---

[1]Our numerical procedure for positioning initial vortex dipoles combined with the finite grid resolution leads to a small variation in the dipole distances and angles with respect to our chosen input parameters. For the dipole distance, the variation is less than $0.1\xi$; for the angle of incidence the variation is kept below 0.1 degree. In our figures we thus report values for $d$ and $\theta$ to this level of accuracy.

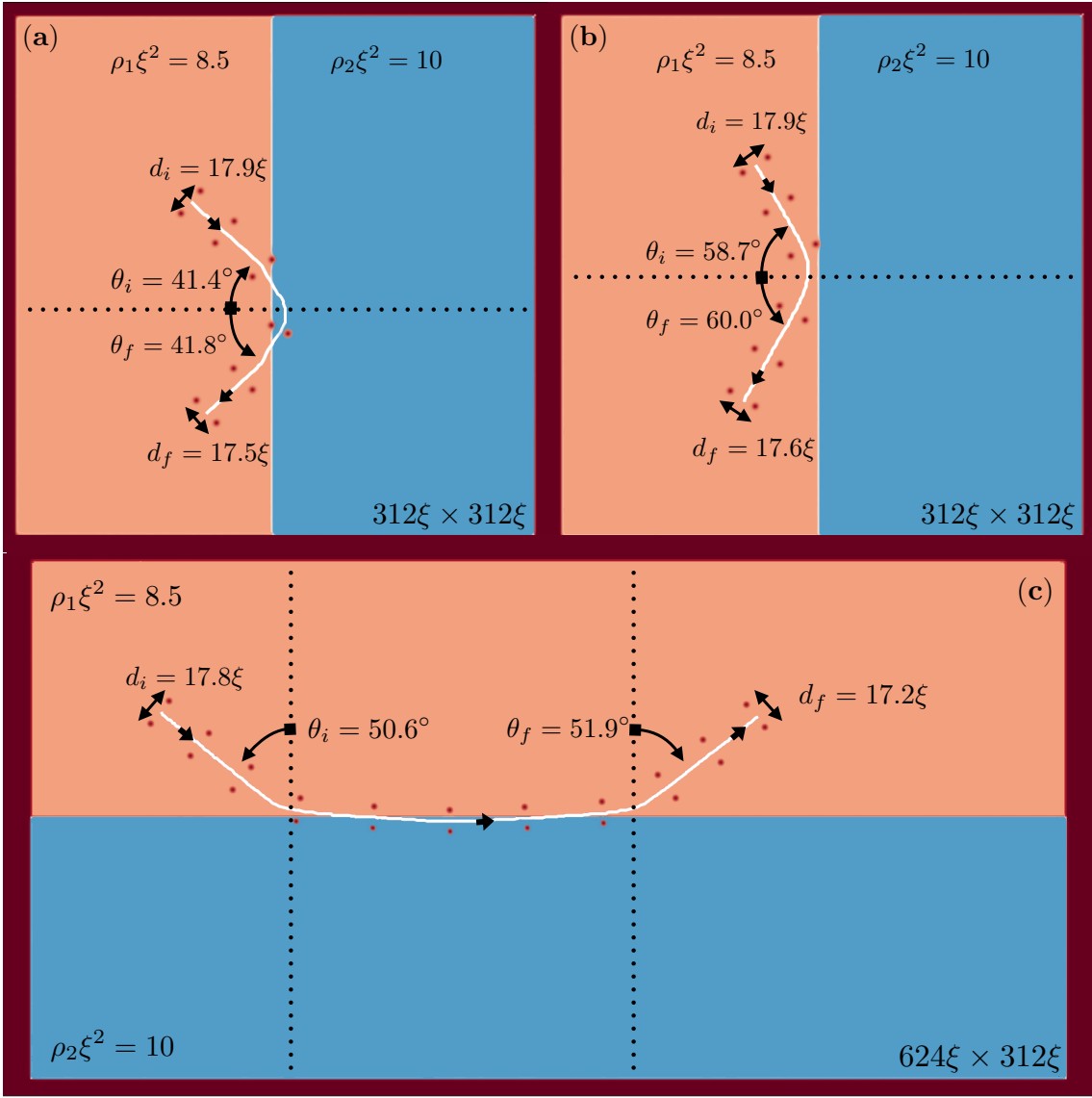

Figure 2: Vortex dipole motion across a density interface with $\rho_1 < \rho_2$. Each subfigure shows the same vortex dipole, superimposed at different times during its motion. The line between the vortices traces out the centre of the dipole; for each case, the critical angle is $\theta_c \simeq 39.7°$. (a) For $\theta_i \gtrsim \theta_c$, the dipole undergoes total internal reflection. (b) For $\theta_i \gg \theta_c$, the dipole again undergoes total internal reflection. (c) For an intermediate range of angles above the critical angle, the dipole straddles the interface between the two regions of superfluid density before undergoing total internal reflection.

3. **Large:** $\rho_1/\rho_2 = 8.5/10$.

For our first study of density-step dependence, we use a single dipole distance for each step, given by $d_i = (11.9\xi, 13.9\xi, 17.9\xi)$ respectively. The values of $d_i$ are chosen in this case to ensure $d_f \approx 10\xi$ upon crossing the interface. This choice ensures that the dipoles have the same energy (both initial and final states) while remaining in the point-vortex regime throughout, and minimising computational time.

In Figure 3 we see the measurements of the incident and refracted angles of the dipoles for each of the three step sizes. The green line in each subfigure represents the expected

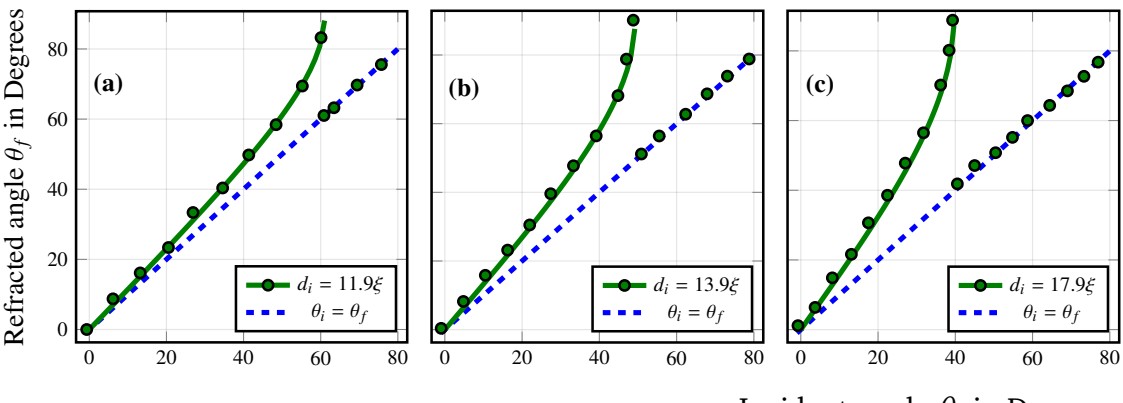

Figure 3: Incident vs refracted angles of low energy dipoles across a small, medium and large change in density ($\rho_1 < \rho_2$). The initial dipole separation distance $d_i$ is in units of $\xi$. A data point on the dashed blue line represents a dipole undergoing total internal reflection, with the incident angle equal to the refracted angle. The green line is the expected result from using the formula 19a. Each subfigure shows the incident vs refracted angle for the: (**a**) small step with density change $\rho_1/\rho_2 = 9.5/10$, (**b**) medium step with density change $\rho_1/\rho_2 = 9/10$, and (**c**) large step with density change $\rho_1/\rho_2 = 8.5/10$.

analytic result from equation (19a) and the green dots represent the data of the incident low energy dipoles taken from simulations. A data point on the $\theta_i = \theta_f$ line represents a dipole undergoing total internal reflection. In Figure 3 (**a**) the refracted angles for the small step show a gradual growth in the refracted angle as the incident angle is increased, and then an abrupt change to reflection at approximately $\theta_i \approx 60°$, close to the analytic prediction of $\theta_c = 61.1°$. In Figure 3 (**b**), the initial dipole separation has been increased to $d_i = 13.9\xi$ to have the dipole energy the same as in Figure 3 (**a**). Dipoles traversing the medium step show a quicker growth in $\theta_f$ as $\theta_i$ increases and then transition to reflection at an angle of approximately $\theta_i \approx 50°$, close to the analytic result of $\theta_c = 49.2°$. For the large step, Figure 3 (**c**) shows that the refracted angle significantly deviates from the $\theta_i = \theta_f$ line as $\theta_i$ increases. The large step has the smallest critical angle, with dipoles reflecting after approximately $\theta_i \approx 40°$, still in reasonable agreement with the analytic prediction $\theta_c = 39.7°$.

To investigate the effect of changing the dipole separation distance (dipole energy) while keeping the potential step the same, we choose the external potential with the large step ($\rho_1/\rho_2 = 8.5/10$) and three different initial dipole separations $d_i = (17.9, 23.9, 30.0)\xi$. The expected critical angles are then $\theta_c = (39.7°, 37.7°, 36.2°)$ respectively. The results are presented in Figure 4, in which the green, orange and red data points represent low, medium and high energy dipoles respectively. We see that for a given system size, a larger change in $\theta_i$ and $\theta_c$ can be effected by varying the potential step/density ratio as opposed to changing $d_i$. This behaviour is due to the fact that the dipole energy (15) depends only logarithmically on the dipole separation distance, whereas the dependence on the background superfluid density is linear.

## 4.4 Anomalous Snell's Law: Interface-Capture

We now examine the dipole motion near the critical angle in greater detail, using the elongated external potential with the large step to find a ground state solution of the GPE. Low energy dipoles ($d_i = 17.9\xi$) were imprinted at 84 evenly spaced incident angles ranging from

$\theta_i = 37.25°$ to $\theta_i = 58.0°$. The critical angle for this system is $\theta_c = 39.7°$.

To obtain better accuracy in initial dipole parameters during the imprinting step, the density profile of a single vortex core was solved numerically using Chebyshev polynomials and then interpolated onto the numerical grid at the desired location for each vortex in the dipole. The phase, Eq. (8), was then imprinted onto the wavefunction at the vortex locations. There was no subsequent imaginary time evolution after the imprinting step.

As the incident angle of the dipoles is increased beyond the critical angle, the dynamics pass through several regimes involving varying levels of transient surface interaction. We classify the trajectories into five separate regimes. Figure 5 shows the trajectories of the centres of the dipoles. The figure is split into two parts for clarity, as the outgoing dipole trajectories pass over one another in distinct regimes. We list here the regimes in order from 1 to 5, and their corresponding colour labels used in Figure 5.

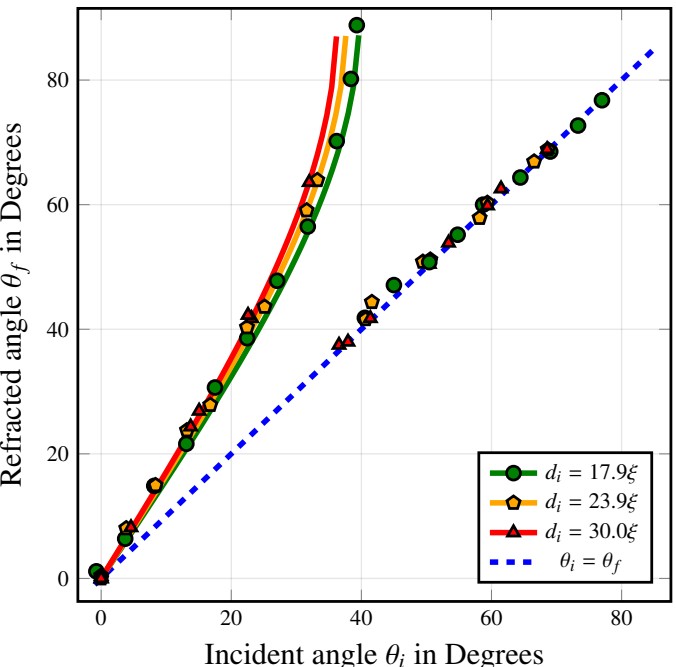

Figure 4: Incident vs refracted angle across the large step ($\rho_1/\rho_2 = 8.5/10$) with low ($d_i = 17.9\xi$), medium ($d_i = 23.9\xi$) and high ($d_i = 30.0\xi$) energy dipoles. Data points on the dashed blue line represent a dipole undergoing total internal reflection, with the incident angle equal to the refracted angle. The solid lines are the predictions of Eq. (19a) for each initial dipole separation distance .

1. **Green:** $\theta_i \in [37.25°, 39.0°]$, $\theta_f \in [75.51°, 85.60°]$
The first 8 trajectories are shown in Figure 5 (**a**). In this first regime, all dipoles undergo ordinary ray-like refraction from a single point of incidence at the interface. Here $\theta_f$ increases with $\theta_i$ until $\theta_i$ nears the analytic critical angle $\theta_c$.

2. **Yellow:** $\theta_i \in [39.25°, 41.75°]$, $\theta_f \in [89.46°, 41.90°]$
In the second regime, $\theta_i$ starts just below the analytic critical angle. Both vortices are transmitted across the interface and propagate parallel to the interface. A further increase of $\theta_i$ causes reflection after an interval of propagation parallel to the interface. This transient surface capture length decreases with increasing $\theta_i$, until the reflection becomes ray-like once again, devoid of any surface-capture dynamics.

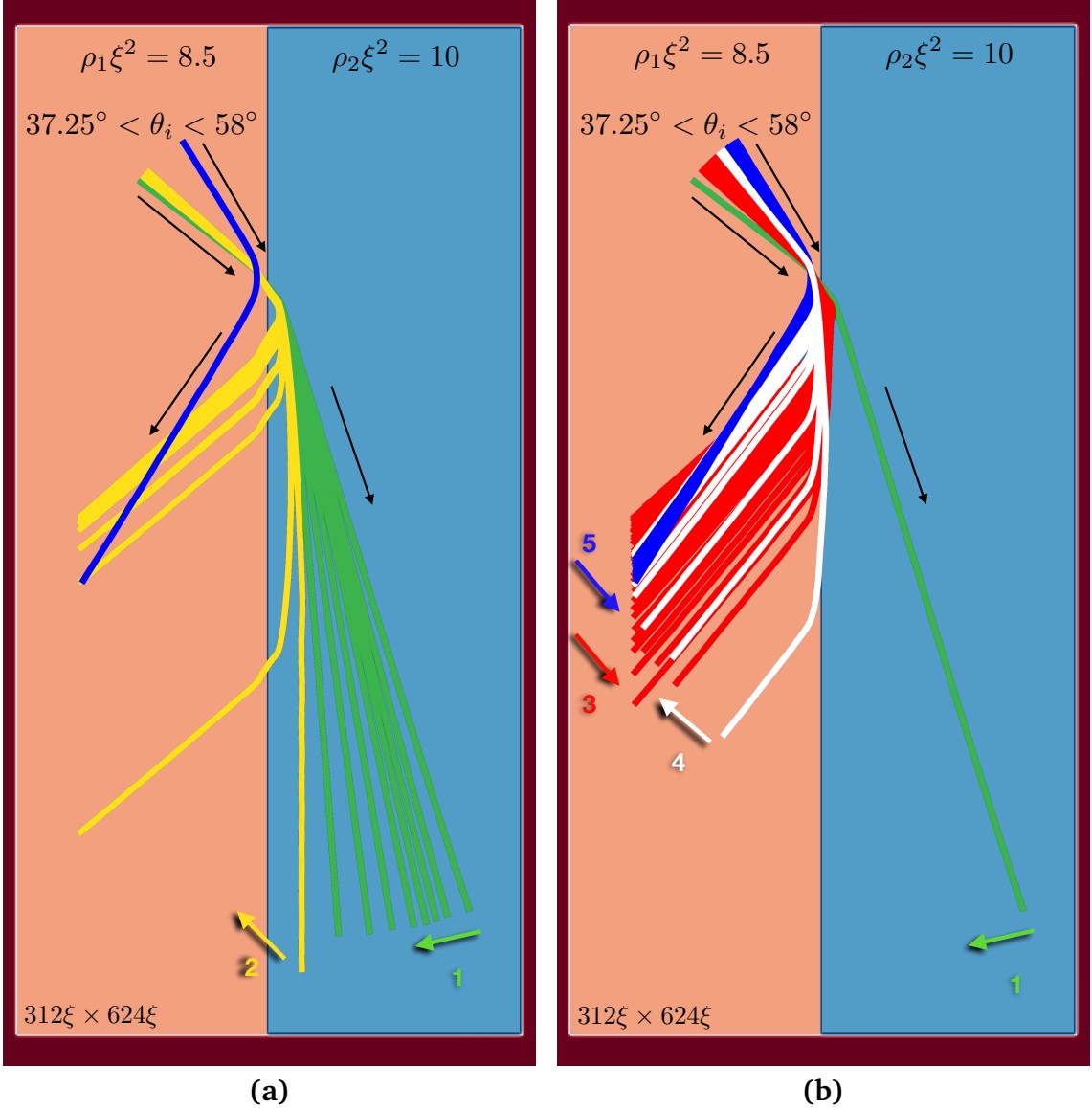

**(a)**                    **(b)**

Figure 5: The trajectories of low energy dipoles ($d_i = 17.9\xi$) across the large step ($\rho_1/\rho_2 = 8.5/10$) within a range of incident angles $\theta_i$ extending from $37.25°$ to $58.0°$. Here the critical angle is $\theta_c = 39.7°$. The figure is divided into two parts for clarity, with each part containing the same bounding trajectories defining the angular range. The coloured arrows point from the outgoing trajectory with the smallest incident angle towards the outgoing trajectory with the largest incident angle for each regime, and the black arrows indicate the direction of dipole propagation. **(a)**: The first regime of behaviour is shown by the green dipole trajectories. In this regime the dipoles undergo refraction. Their incident angles are all below the critical angle, and their outgoing angle increases as a function of the incident angle. The second regime is shown by the yellow trajectories, beginning with $\theta_i = 39.25°$. **(b)**: The third and fourth regimes are given by the red and white dipole trajectories respectively. The fifth regime is given by the blue dipole trajectories beginning with $\theta_i = 54.25°$. See text for a description the regimes.

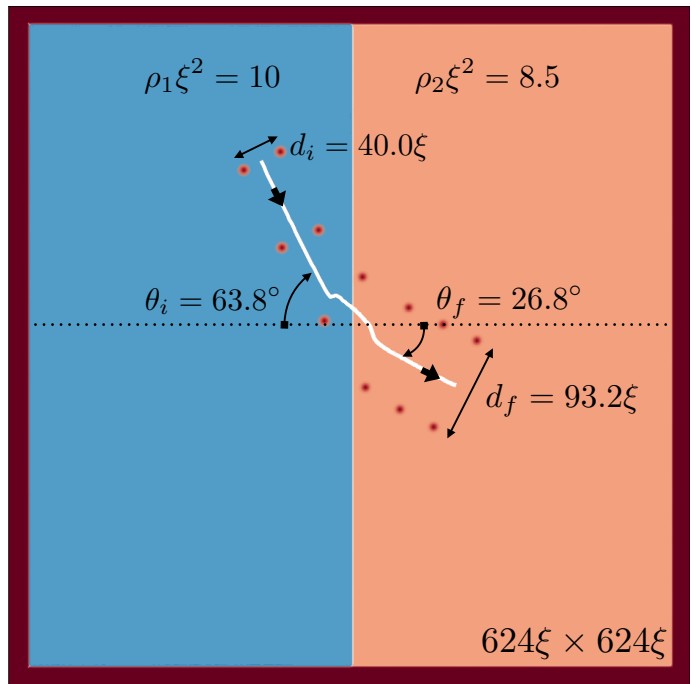

Figure 6: A high energy ($d_i = 40.0\xi$) dipole is incident upon the large step ($\rho_1/\rho_2 = 10/8.5$) at a high initial angle ($\theta_i = 63.8°$). The dipole undergoes refraction, leaving the interface at a smaller outgoing angle.

3. **Red:** $\theta_i \in [42.0°, 50.25°]$, $\theta_f \in [42.17°, 50.92°]$
   In the third regime [Figure 5 (b)], the dipoles initially undergo ray-like reflection at the interface. As $\theta_i$ increases, the surface capture length monotonically increases. The nature of surface capture is fundamentally different from regime 1 (Yellow), in that now the dipole is straddling the interface. This is consistent with the expectation that greater momentum normal to the interface would lead to a more complete interface crossing. The surface-capture length eventually reaches a maximum extent for this regime, defining the regime boundary.

4. **White:** $\theta_i \in [50.50°, 54.25°]$, $\theta_f \in [51.91°, 54.59°]$
   In the fourth regime, the dipoles again reduce their surface-capture length as $\theta_i$ increases. This is a similar behaviour as seen for regime 3 (Red). When the surface-capture length reaches zero, the next regime begins.

5. **Blue:** $\theta_i \in [54.50°, 58.0°]$, $\theta_f \in [54.89°, 58.34°]$
   In the fifth regime, the dipoles undergo ray-like total internal reflection when reaching the interface, with neither vortex crossing the interface.

All dipole trajectories were observed to obey Snell's law. For ray-like propagation, the angle relationship is straightforward to verify. For the anomalous surface-capture regime, Snell's law was found to describe the asymptotic states when measuring the outgoing angle of the trajectory about its last point of contact with the interface.

## 4.5 High- to Low-Density Trajectories

Finally, we consider the case where the dipole undergoes refraction from a high to a low-density region of superfluid ($\rho_1 > \rho_2$). The external potential with the large step

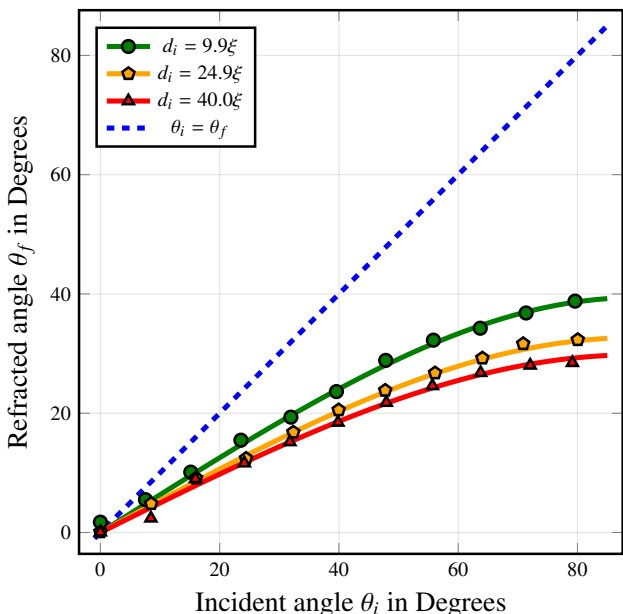

Figure 7: Incident vs refracted angle across the large step ($\rho_1/\rho_2 = 10/8.5$) with low ($d_i = 9.9\xi$), medium ($d_i = 24.9\xi$) and high ($d_i = 40.0\xi$) energy dipoles. Data points on the dashed blue line represent a dipole undergoing total internal reflection, with the incident angle equal to the refracted angle. The solid lines are the predictions of Eq. (19a) for each initial dipole separation distance.

($\rho_1/\rho_2 = 10/8.5$) was used with low, medium, and high energy dipoles with separation distance $d_i = (9.9, 24.9, 40.0)\xi$ respectively. In order to reduce boundary interactions for the medium and high energy dipoles, a larger system size was used with area $624\xi \times 624\xi$. The $\theta_i$ ranged from zero to $80°$ in steps of $8°$. A representative example is shown in Figure 6, where a high energy dipole ($d_i = 40.0\xi$) incident upon the interface at a large angle undergoes refraction and leaves the boundary at a smaller outgoing angle. No critical angle is observed for dipoles traversing from high to low-density regions, as such dipoles always undergo refraction. Figure 7 shows the change in $\theta_f$ due to a change in $d_i$, for the large step. The green, orange and red data points represent the low, medium and high energy dipoles respectively. We see that $\theta_f$ increases at a slower rate than $\theta_i$. The effect of changing the dipole energy is readily seen here, with dipoles of high energy showing a stronger divergence from the $\theta_i = \theta_f$ line than dipoles with lower energy.

## 5 Discussion and Conclusion

In this section, we make a connection with the optical concept of a refractive index, discuss interface physics, and present our conclusions.

### 5.1 Refractive index

In a compressible superfluid with healing length $\xi$ the JRS forms when the vortices approach within a distance $\lambda\xi$, where the constant $\lambda \sim 2.3$ [31], but the precise value of $\lambda$ is independent of the background density of the superfluid. Hence we can define the JRS reference

speeds on each side of the interface as

$$C_{JRS,i} = \frac{\hbar}{m} \frac{1}{\lambda \xi_j} \qquad \text{where} \quad j = i, f, \tag{22}$$

and where $\lambda$ is the *same* constant on either side. We then introduce the index of refraction for a vortex dipole as

$$n_j \equiv \frac{C_{JRS,j}}{v_j} = \frac{\hbar}{m} \frac{1}{\lambda \xi_j} \frac{m d_j}{\hbar} = \frac{d_j}{\lambda \xi_j} = \frac{1}{\lambda \alpha} \exp\left( \frac{E_j}{2\pi \rho_j} \frac{m}{\hbar^2} \right), \tag{23}$$

where $v_j = \hbar/(m d_j)$ is the velocity of a vortex dipole. In terms of $c_j \equiv \sqrt{g_{2D} \rho_j / m}$, the local speed of sound, Eq. (19a) can now be written as

$$\frac{\sin(\theta_i)}{\sin(\theta_f)} = \frac{d_f \rho_f}{d_i \rho_i} = \frac{v_i \rho_f}{v_f \rho_i} = \frac{c_f n_f}{c_i n_i}, \tag{24}$$

where the common JRS scale factor $\lambda$ does not play an obvious role as the law can be phrased in terms of the two sound speeds. This formulation allows the assignment of a particular refractive index for a given region of superfluid. However, the result is not as simple as for optics, because there is no well-defined reference speed that is material independent.

## 5.2 Interface Physics

In contrast to ordinary ray optics, Snell's law for the quantum vortex dipole is dependent on the local speed of sound, and the natural reference speed is determined by the closest distance of approach for the vortices — the scale where the vortices lose their topological character and form a Jones-Roberts soliton. While many features of ray optics are observed, including total internal reflection, there are clear departures from ray optics when the internal structure of the dipole becomes important. Most notably, due to transient capture by the interface, the ordinary ray-like propagation breaks down near the critical angle for total internal reflection. However, a detailed study of interface dynamics requires systematic exploration of several parameters: dipole separation, density change across the interface, and the shape of the interface potential; such a systematic study is beyond the scope of this work, and the question of what precisely occurs during the transient surface capture remains an interesting open question. In particular, it would be interesting to learn why surface capture times can be so long, and what changes when the dipole is released. There may also be a significant change in compressible energy or the shape of the interface during the process. It would be interesting to investigate the possibility of indefinite capture, with potential for solitonic behaviour at the interface. Finally, there may be further useful analogies between the transient surface-capture phenomenon and the Goos-Hänchen effect [38] or to surface-plasmon physics.

## 5.3 Conclusion

Snell's law is a universal governing principle of ray optics, describing the incidence of an optical ray on a translationally invariant interface between two uniform optical media. More generally any excitation undergoing rectilinear propagation at an interface can be expected to obey an analogue Snell's law due to the conservation of the component of momentum parallel with the interface. Our results show that this principle governs the motion of a quantum vortex dipole moving in a planar superfluid. The finite vortex core size in a Bose-Einstein condensate introduces an additional energy change when the vortices cross the interface, and this effect must be included to obtain the correct formulation of Snell's law for a quantum vortex dipole.

Despite finite-size effects at the interface, Snell's law governing the relationship between incoming and outgoing asymptotic states is always satisfied. Explorations of the optical analogy may yet suggest ways to create simple optical elements such as lenses and mirrors for quantum vortex dipoles and may reveal additional mechanisms for manipulating vortex motion [17, 28].

## Acknowledgements

We thank M. T. Reeves and A. L. Fetter for useful discussions.

**Funding information** ASB was supported by the Marsden Fund (Contract UOO1726), and BPA was supported by the National Science Foundation (Grant Number PHY-1607243). MMC, XY, and ASB acknowledge support from the Dodd-Walls Centre for Photonic and Quantum Technologies.

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
