# Peer review of "Snell's Law for a vortex dipole in a Bose-Einstein condensate"

_SciPost Physics, doi:SciPost Phys. 6, 032 (2019)_

## Round 1 · Referee Report · Anonymous · 2018-12-4

Strengths

1- Provides a useful guide for the behavior of vortex dipoles (which are self-propelled localized excitations) as they pass through interfaces of sharply changing density.

2- A thorough numerical investigation with some analytic insight

Weaknesses

The text could be streamlined and clarified in a few places. But this is only a very minor issue and didn’t really slow my reading to any significant degree.

Report

The manuscript by Cawte et al. studies the motion of vortex dipoles as they cross a step interface between regions of differing density. The system is in a quasi-2D square well configuration and the different densities are created by a step in the potential along the bottom of the well. They numerically analyze the reflection/refraction versus incident angles and compare it to a Snell’s law-like analytical prediction. They find that this prediction is obeyed well before and long after the interaction with the surface, even when for some incident angles there can be complicated intermediate behavior such as surface capture.

I recommend this manuscript for publication in SciPost after some minor revisions. In my view this work is interesting and could guide future experiments on vortices in flattened geometries. The numerical study is thorough, while at the same time being clear and easy to read.

Requested changes

1- The derivation of vortex dipole momentum [eq (11)] is not easy to follow. Why is the 2D integral reduced to a pair of line integrals along the top and bottom borders of the system? Also, why are the limits of the integral only from -d/2 to d/2. I would have expected nonzero contributions to $P_y$ for $x$ values outside of these limits too. Some further explanation in the text would be beneficial.

2- In Eq. (19b) does $\xi_f = \xi_i$ for the present work? Shouldn’t this always be the case when in equilibrium? A comment on this would be useful since sometimes people refer to local effective chemical potentials that depend on the local potential.

3- Do the authors know what is the reason for long surface capture times? What is changing during the capture time that finally allows a dipole to be released? Has it something to do with the reorganization of superfluid flow fields at short to moderate distances? Do you think there are any conditions that could cause a dipole to be captured indefinitely? Discuss.

4- It looks like a couple of typos on the second line of section 4.5. Shouldn’t it read $\rho_i>\rho_f$ and $\rho_2/\rho_1 =$ … ?

---

## Round 1 · Referee Report · Anonymous · 2018-12-21

Strengths

The paper is well written and the results are very clear and interesting. The agreement between the theory and numerical results are excellent, indicating that the proposed “Snell’s law” for vortex dipoles is reasonable.

Weaknesses

Some corrections are needed (written in the report).

Report

The authors of the present paper investigate the dynamics of a vortex dipole incident upon the interface of superfluid with different densities. They derive the relation between the incident and refraction angles, as Snell’s law in optics, which is numerically confirmed by solving the Gross-Pitaevskii equation. They also find an interesting intermediate phenomenon, where the incident vortex dipole travels along the interface before it is reflected.

The paper is well written and the results are very clear and interesting. The agreement between the theory and numerical results are excellent, indicating that the proposed “Snell’s law” for vortex dipoles is reasonable. I recommend this paper for publication in SciPost Phyiscs.

Minor comments:
1) Captions in Figs. 4 and 7: Do the solid lines represent Eq. (19)? The caption should state it explicitly.

2) Caption of Fig. 5: The meaning of the colored arrows should be written.

3) In Fig. 5(b), the traveling distance of the vortex dipole along the interface increases with the incident angle for the red lines, whereas it decreases for the yellow and white lines. Is there some explanation for this behavior?

4) The present results may be related with PRA 85, 023618 (2012), in which the dynamics of vortex dipoles across the interface between binary BECs are studied.

Requested changes

Written in the report.

---

## Round 4 · Author Response

We thank the referees for their considered and constructive comments. We believe that addressing the comments has clarified the paper and strengthened the presentation. We have made a number of changes, as detailed below. We hope that article is now deemed suitable for publication.

---

## Round 4 · List of Changes

First Reviewer comments

1. The derivation of vortex dipole momentum [eq (11)] is not easy to follow. Why is the 2D integral reduced to a pair of line integrals along the top and bottom borders of the system? Also, why are the limits of the integral only from $-d/2$ to $d/2$. I would have expected nonzero contributions to $P_y$ for $x$ values outside of these limits too. Some further explanation in the text would be beneficial.

To address this, we have significantly reworked the beginning of section 2.2 leading up to the dipole momentum, Eq. (12); see revised version for details.

2. In Eq. (19b) does $\xi_f=\xi_i$ for the present work? Shouldn't this always be the case when in equilibrium? A comment on this would be useful since sometimes people refer to local effective chemical potentials that depend on the local potential.

To address this, we have changed in the initial part of the section Snell's Law for vortex dipoles to state:

"We consider a superfluid with densities $\rho_1$ and $\rho_2$ separated by a sharp interface in the density created by an external potential (see Fig. 1). There are two healing lengths dependent on the local chemical potential $\xi_j = \hbar/\sqrt{m g_{2D}\rho_j}$ for $j=1,2$. Hereafter, we work in units of the healing length $\xi=\textrm{min}(\xi_1,\xi_2)$, the shortest length scale set by the highest particle density. To distinguish the static background densities $\rho_j$ from dynamical properties of the vortex dipole, we use the subscripts $(i,f)$ to refer to initial and final states of the dipole respectively. The initial or final state may refer to a dipole in either region $\rho_1$ or $\rho_2$ of superfluid density, as will be made clear from the context."

3. Do the authors know what is the reason for long surface capture times? What is changing during the capture time that finally allows a dipole to be released? Has it something to do with the reorganization of superfluid flow fields at short to moderate distances? Do you think there are any conditions that could cause a dipole to be captured indefinitely? Discuss.

To address these questions, the conclusion section has been split up into subsections, with a new interface physics subsection added. The contents of that section now read:

"Most notably, due to transient capture by the interface, the ordinary ray-like propagation breaks down near the critical angle for total internal reflection. However, a detailed study of interface dynamics requires systematic exploration of several parameters: dipole separation, density change across the interface, and the shape of the interface potential; such a systematic study is beyond the scope of this work, and the question of what precisely occurs during the transient surface capture remains an interesting open question. In particular, it would be interesting to learn why surface capture times can be so long, and what changes when the dipole is released. There may also be a significant change in compressible energy or the shape of the interface during the process. It would be interesting to investigate the possibility of indefinite capture, with potential for solitonic behaviour at the interface. Finally, there may be further useful analogies between the transient surface-capture phenomenon and the Goos-Hänchen effect [34] or to surface-plasmon physics."

4. It looks like a couple of typos on the second line of section 4.5. Shouldn't it read $\rho_i > \rho_f$ and $\rho_1/\rho_2= 10/8.5$

This issue has been fixed.

Second Reviewer comments

1. Captions in Figs. 4 and 7: Do the solid lines represent Eq. (19)? The caption should state it explicitly.

Figure 4 and 7 have had the following sentences added to the caption:

"Data points on the dashed blue line represent a dipole undergoing total internal reflection, with the incident angle equal to the refracted angle. The solid lines are the predictions of Eq. (19a) for each initial dipole separation distance."

2. Caption of Fig. 5: The meaning of the colored arrows should be written.

The following sentence was added to the caption:

"The coloured arrows point from the outgoing trajectory with the smallest incident angle towards the outgoing trajectory with the largest incident angle for each regime, and the black arrows indicate the direction of dipole propagation."

3. In Fig. 5(b), the traveling distance of the vortex dipole along the interface increases with the incident angle for the red lines, whereas it decreases for the yellow and white lines. Is there some explanation for this behavior?

To address these questions, the conclusion section has been split up into subsections, with a new interface physics subsection added. See suggestion 3 raised by the first reviewer.

4. The present results may be related with PRA 85, 023618 (2012), in which the dynamics of vortex dipoles across the interface between binary BECs are studied.

A new paragraph about this work has been added to the introduction, and the start of the succeeding paragraph has been changed. The relevant part of the introduction now reads:

"In recent numerical work [30], it was shown that a vortex dipole at sufficiently high velocity could cross an interface in an immiscible two-component BEC. At lower velocities, the dipole either disappeared or disintegrated, with the remnants moving along the interface. For vortex dipoles with an oblique angle of incidence upon the interface, the cores of the dipole were found to be asymmetrically filled with the first component of the BEC after propagation into the second component. In general, the dipole-interface interaction was found to be quite complex.

In this work, we study the motion of a single vortex dipole incident on a density-step interface in a single-component BEC. The interface is created by adding an abrupt potential step, resulting in two large regions of different potential and density."

Additional changes

1. Just after equation 1, where the Jones-Roberts soliton has been introduced, the sentence:

"... this speed is a less than the speed of sound in each region of the system and sets a natural boundary to the parameters of the dipoles we study, in particular introducing the smallest dipole moment that we may consider."

has been changed to

"...this speed is less than the speed of sound in each region of the system and sets a natural boundary to the parameters of the dipoles we study, in particular setting the scale of the smallest dipoles that we may consider."

2. Right above Eq 5, it has been made clearer that the $\psi_0(z)$ must be normalized to N for $g_{2D}$ to have this form.

3. The dimensionless equations used for numerical simulation of the system have been dropped from section 2.1.

4. At the end of the sentence that contains Eq 7, the sentence "and $r$ is the distance from the center of the core." has been added.

5. After Eq 17 the following sentence has been added:

"We emphasize that in contrast with intuition based on the Newtonian mechanics of classical particles, the dipole energy and momentum scale with $d$: larger $d$ gives larger dipole energy and momentum, but smaller dipole velocity. "

6. The term "the wedge" has been changed throughout the text to "range of incident angles".

7. A reference for the Goos-Hänchen effect has been added.

Minor changes

1. The trap dimensions on the figures has been changed from $x \times y \xi$ to $x \xi \times y \xi$.

2. The last line of the abstract: the word "surface" has been changed to "interface".

3. In the first paragraph of the intro, the word "and" was added after "confined by parabolic potentials,". At the end of the sentence "In a uniform BEC...", the phrase "its separation" was changed to "the distance between the vortices". After the sentence "superfluid with inhomogeneous density" the word "step" was added.

4. Added "the" before "phase velocity" in the line below Eq 1.

5. Removed an extra "a" in the line before "less than the speed" in the Jones-Roberts soliton paragraph.

6. In the sentence below Eq 12, "inserting" was used instead of "plugging". The term "two-dimensional" was added before "expression".

7. Equation 3 was changed from $H$ to $E$ to be consistent with equation 14.

8. Page 7, Second line from end: "riches" was corrected to "reaches".

9. "Polynomials" after Chebyshev is no longer capitalized.

10. Fig 4 caption. Line 3: "represents" was changed to "represent".

You are currently on this page

Resubmission 1811.02110v4 on 26 February 2019

---

## Editorial Decision

published